

# A risk assessment methodology to evaluate the risk failure of Managed Aquifer Recharge in Mediterranean basin

Paula Rodríguez-Escales[1,2], Arnau Canelles[1,2], Xavier Sanchez-Vila[1,2], Albert Folch[1,2], Daniel Kurtzman[3], Rudy Rossetto[4], Enrique Fernández-Escalante[5], João-Paulo Lobo-Ferreira[6], Manuel Sapiano[7], Jon San-Sebastián[5], and Christopher Schuth[8]

[1]Dept. of Civil and Environmental Engineering. Universitat Politècnica de Catalunya, Barcelona, Spain
[2]Associated Unit: Hydrogeology Group (UPC-CSIC)
[3]Institute of Soil, Water and Environmental Sciences, The Volcani Center, Agricultural Research Organization, Rishon LeZion, Israel
[4]Institute of Life Sciences, Scuola Superiore Sant'Anna, Pisa, Italy
[5]Empresa de Transformación Agraria (TRAGSA), RD dpt., Madrid, Spain
[6]Laboratorio Nacional de Engenharia Civil, Lisboa, Portugal
[7]Energy and Water Agency, Luqa, Malta
[8]Institute of Applied Geosciences, Darmstadt University of Technology, Darmstadt, Germany

**Correspondence:** Paula Rodríguez-Escales (paula.rodriguez.escales@upc.edu)

**Abstract.** Managed Aquifer Recharge (MAR) can be affected by many risks. Those risks are related to different aspects of recharge, like water availability, quality, civil engineering issues, etc. Many other works have acknowledged risks of this nature, theoretically, however their quantification and definition has not been developed. In this study, the risk definition and quantification has been performed by means of Fault Trees and Probabilistic Risk Assessment (PRA). We defined a fault tree

5 with 65 basic events applicable to operation phase. After that, we have applied this methodology to six different Managed Aquifer Recharge sites located at the Mediterranean Basin (Portugal, Spain, Italy, Malta and Israel). The probabilities of the basic events were defined by expert criteria, based on the knowledge of the different managers of the facilities. From that, we conclude that in all sites, the perception of the expert criteria of the non-technical aspects were as much or even more important than the technical ones. Regarding the risk results, we observe that the total risk in three of six sites was equal or above 0.90.

10 That would mean that the MAR facilities have a risk of failure equal or higher than 90% in the period of 2-6 years. The other three sites presented lower risks (75%, 29% and 18% respectively).

*Copyright statement.* TEXT

# 1 Introduction

Water scarcity, the chronically lack of sufficient quality water to supply a specific area, is one of the major global challenges. In
15 the Mediterranean Basin, due to low overall precipitation and a pronounced irregularity of rainfall events, it has direct impacts on economic sectors that depend on water, such as agriculture, tourism, and related industries (Fader et al., 2016; Maliva and



Missimer, 2012; Navarro-Ortega et al., 2012; Stanhill et al., 2015). Besides this, the population in the Mediterranean Area increased from 81 million in 1960 to 145 million in 2011 (European Environment Agency, 2015), placing additional stress upon existing water resources. Moreover, the Mediterranean Basin is one of the most sensitive regions of the world to climate changes resulting from human activities; according to the latest IPCC projections, average precipitation could decrease by more

than 10%, with a larger decrease in summer and in the southern areas (Pachauri, R.K., Allen, M.R., Barros, V.R., Broome, J., Cramer, W., Christ, R., Church, J.A., Clarke, L., Dahe, Q., Dasgupta, P. and Dubash, 2014).

At the same time, large water quantities are lost to the Mediterranean Sea as surface runoff and discharges from rivers, treated and untreated wastewater, or excess water from various sources during periods of low demand. These alternative water sources potentially can help to increase water availability, both in general terms and in periods of high demand, therefore improving

water security. The main factors hindering the effective use of such waters are related to concerns about water quality and the lack of sufficient low cost intermediate storage options. In principal, large storage capacity is available in shallow aquifers, mostly in thick unsaturated zones or in already depleted overexploited aquifers. Managed Aquifer Recharge (MAR) takes advantage of this available storage.

MAR is defined as the intentional infiltration of water into aquifers with the purpose of either later recovering that water for

different uses (agricultural, industrial or urban), or obtaining an environmental benefit (Dillon et al., 2009). MAR includes a range of recharge options (surface or subsurface) and water sources (natural, reclaimed or desalinated) (Bouwer, 2002; Dillon, 2005; Dillon et al., 2009; Maliva and Missimer, 2012; Sprenger et al., 2017). Furthermore, MAR can involve different engineering solutions, among them, infiltration ponds, surface spreading, bank filtration, and wells infiltrating into either the unsaturated or the saturated zones. In addition, water quality can be improved through MAR due to the combination of chem-

ical and biological reactions during transport of the infiltrated water. Water can either be recovered at the point of infiltration (ASR – Aquifer Storage and Recovery), or some distance downgradient (ASTR – Aquifer Storage, Transport and Recovery). The infiltrated water can enable hydraulic control of an aquifer, e.g. to prevent seawater intrusion, aid aquifer quality recovery (amelioration of the groundwater quality), or protect surface water bodies such as wetlands or marshes. Altogether, MAR links water reclamation, water reuse and water resources management.

Due to these beneficial effects, MAR is now widely regarded as a useful tool to ensure a safe and good quality water source for the increasing demand. However, to guarantee the success of any MAR project, some essential elements need to be considered (Dillon et al., 2009), (i) an adequate source of water for recharge, (ii) a suitable aquifer to store and recover water, (iii) available land to construct the facilities, (iv) a sufficient demand for the recovered water, and (v) the capability to efficiently manage such a project. If any of these elements fails, a MAR project is usually not viable. However, the listed factors seem to involve

infrastructural and management aspects only, and ignore legal, social, economic, and political constraints that can significantly entangle MAR application, eventually leading to failure of the project as a whole. In addition, the analysis of the potential success of a MAR project should account both for the initial set-up of installation, and also for the potential issues that will arise during its operation (European Community, 2000).

The most common identified technical risks for MAR facilities (e.g. Asano and Cotruvo, 2004; Gale et al., 2006; Leviston

et al., 2006; Maliva and Missimer, 2012) are those related to: (i) the operation of the facility (low recovery rates, clogging, me-





chanical/structural damage, low storage efficiency, high energy consumption); (ii) water quality, either recharged or extracted; (iii) hydraulic engineering impacts, such as rock fracturing, subsidence, or host porous media dissolution; and (iv) environmental impacts, including reduced water outflow to springs and rivers, proliferation of pests and odors, and impact on aquifer dependent ecosystems.

MAR facilities can also be affected by legal, social, economic, and political issues, which increase the risk of failure, meaning that the facility would not begin, or continue operation. Therefore, a full and complete risk assessment must encapsulate all relevant constraints and their confidence level, at a given time and projected into the future. In addition, risk evaluations might include the implementation of measures to control risk, by either diminishing the probability of occurrence of a given hazard, or reducing/correcting its effects if they eventually occur. The implementation of a MAR facility is therefore subject

to a relatively high degree of uncertainty (Bouwer, 2003; Dillon et al., 2009). Uncertainty can be managed using Probabilistic Risk Assessment (PRA), a concept used in various fields of science and engineering. Risk is defined here as the probability of an undesired outcome to happen (evaluated in terms of percentage of occurrence, return period, etc.) and an evaluation of the potential damage that a particular outcome might cause (amount of damage, adverse health effects, impact to ecosystems, etc.). Different definitions for risk in MAR are available in the literature; Maliva and Missimer (2012) defined it as the feasibility

(technical and economic) to meet regulatory requirements for aquifer recharge.

Several methods are available for risk evaluation. One such method is the development of Fault Trees, already used in engineered systems (Bedford, T. and Cooke, 2003; Vesely et al., 1981). Since MAR systems comprise a mixture of natural and engineered components, this approach has received some attention in the hydrological community (e.g. Bolster et al., 2009). The basic idea of PRA based on Fault Trees (PRA-FT) is to take a complex system, difficult to be handled as a whole, and

to divide it into a series of quasi-independent simpler events that are manageable individually (i.e. basic events). Once probabilities of occurrence of basic events are computed, they are recombined in a systematic manner to provide the overall risk assessment of the system as a whole. Examples of applications of PRA-FT in hydrogeology include De Barros et al. (2011, 2013) and Jurado et al. (2012).

Although some approaches to evaluate the risk of a MAR system have been developed (Assmuth et al., 2016; Ayuso-Gabella

et al., 2011; Dillon et al., 2016; Ji and Lee, 2017, 2016; Juntunen et al., 2017; Page et al., 2010; Toze et al., 2010), comprehensive studies that integrate both non-technical and technical factors are absent (Nandha et al., 2015). In this study, we (i) present precisely an integrated PRA-FT that is applicable for a general MAR facility, and (ii) apply it to six different MAR facilities, that were part of the EU FP7 project MARSOL, located in five different Mediterranean countries: Portugal (1), Spain (2), Italy (1), Malta (1), and Israel (1). To achieve these goals, first, basic events that can lead to MAR failure were compiled

based on a literature review of 51 MAR facilities worldwide, and on data from the MARSOL project. The next step was the development of six individual fault trees for the test sites, and assigning probabilities of occurrence for these events. Finally, we used the six sites to compare the different realities, and to test the relative relevance of technical versus non-technical events.





## 2 Literature review: events involved in MAR failure

MAR failure is defined as the need to stop operation of the facility. This failure can be either complete or partial, partial failure means that it is possible to mitigate the problem in a short period of time, so that the facility can be put back to operation, where complete failure implies that the facility needs to undertake significant changes and reparations in order to work again (or even not working ever again after the failure). In this paper, the failures exposed are considered as partial failures due to the fact that none of the MAR facilities reviewed in the literature permanently stopped working after those failures occurred.

Basic events that can lead to MAR failure were compiled based on a literature review on the problems encountered by different facilities around the world (Aiken and Kuniansky, 2002; Alazard et al., 2016; Assmuth et al., 2016; Bhusari et al., 2016; Chaoka et al., 2006; Flint and Ellett, 2005; Masetti et al., 2016; Murray and Ravenscroft, 2010; Petersen and Glotzbach, 2005; Brian J. Schneider, Henry F. H. Ku and Oaksford, 1987; Subbasin et al., 2006; Sultana and Ahmed, 2016; Tredoux et al., 2009; Tredoux and Cain, 2010; Tripathi, 2016). We revised 51 MAR facilities at 47 sites (some sites involved more than one facility) located in different countries and climatic conditions worldwide: Australia, Belgium, Botswana, China, Finland, France, Germany, India, Israel, Italy, Jordan, Namibia, South Africa, Spain, Tunisia, and USA. We classified the facilities according to infiltration typology: deep wells (24), surface infiltration (22), and vadose infiltration (5). A summary of the facilities and details can be found in section S1 of the supplementary material.

We then sorted the main causes of MAR failure in terms of frequency of appearance for deep wells and infiltration basins (Figure 1). Furthermore, we classified these problems into technical and non-technical problems and sub-classified them into different categories. The technical problems with the most occurrence were clogging and the presence of nutrients; they were present in 40-50% of the reviewed facilities (Figure 1) and in all types of MAR facilities. Three types of clogging were reported, being in order of decreasing importance, biological, physical and chemical. On the other hand, the nutrient issues were mainly related to the presence of nitrogen and phosphorus in the recharge water, mostly associated with the use of insufficiently treated reclaimed or surface water, with high nutrients levels, for recharge.

In general terms, quality and infiltration issues were the main aspects that limited the viability of MAR facilities. In fact, the six first technical reasons of MAR failure were the same in all facilities: clogging, nutrients, metals, droughts, low infiltration rate and salinity-sodicity. This can be explained because MAR facilities are often in semi-arid countries were droughts (a main problem from the quantitative point of view) are common. Quantity issues were seldom relevant, and only in infiltration basins. Civil work failures and natural hazards were rarely reported as problems.

Regarding the non-technical aspects, they were classified into four groups: legal constraints, economic constraints, social unacceptance, and governance related problems. The actual issues identified in the MAR facilities revised were thus related to cost (maintenance and installation of the MAR facility), legal aspects (mainly sanitary issues for the infiltrated or the reclaimed waters), and local constraints (land permissions and urban planning issues).




# 3 Methodology. Development of the fault trees and risk evaluation.

The methodology used consisted of four steps, modifying the scheme followed by Bedford, T. and Cooke (2003): (1) the definition of the concept of system failure and the identification of the key events that would potentially result in such failure; (2) construction of the fault tree depicting the combination of events, seeking the combination of all possible events that may contribute to system failure (where all events should be as independent from each other as possible); (3) developing a probabilistic representation of the fault tree using Boolean algebra and; (4) computing the individual probabilities of event occurrence using conservative approaches and individual event probabilities and upscaling to calculate the global risk of the facility.

## 3.1 Failure definition and identification of key events

The first step was the identification of the key events that can produce a failure in a general MAR facility by reviewing the literature (Section 2) and, as a second step, by an extension based on the knowledge and the experience of the facility managers. Failure was based on operation stage, which implies the non-properly functioning of the MAR facility, or the cease of its operation for a prolonged time.

## 3.2 Fault tree construction

The eight categories defined before (technical and non-technical) are described by a few key events, giving a total of 65 (21 Technical and 44 Non-Technical) (see Figure 2). A short definition of all the events can be found in Figure 2 and in the S2 of the Supplementary material.

## 3.3 Probabilistic representation of the fault tree

The probabilistic analysis is based on two steps: (1) defining a specific probability for each key event to occur, and (2) combining the different events' probabilities, using Boolean algebra, to assign probabilities to the boxes (events) into the one placed immediately above. In this section we illustrate this approach for simplicity and completeness. Additional details about this methodology can be found in Tartakovsky (2007).

For each event, we specify a number of basic subevents following two models: (1) if any basic subevent occurs, then the event will also occur, thus, equivalent to an "OR" operator in Boolean logic; (2) all basic subevents must occur for the event to take place, characteristic of the "AND" operator. So, denoting E as the event, and $e_i$,i=1,...,n as the basic subevents, the "OR" operators involves:

$$E(or) = U_{1,...,n} e_i \tag{1}$$





while the "AND" operator results in:

$$E(and) = \cap_{1,\dots,n} e_i \tag{2}$$

We illustrate it with a simple example (Figure 3), considering that an issue with either social or political implications increase the probability of having a non-technical MAR failure (which in turn would increase the chance of a general MAR failure).

According to the methodology described, we can obtain the probability of the main event (NT), P[NT], as a function of those of the basic events SO and PO

$$P(NT) = P(SO \cup PO) = P(SO) + P(PO) - P(SO \cap PO) \tag{3}$$

Notice that the last term in (4) indicates the product of an intersection; this would be also the case if instead of an "OR" operator we had an "AND" operator, so that in such a case we would have the following formula, $P(NT) = P(SO \cap PO)$. To compute the probability of the intersection of basic events, we rely on the concept of conditional probability (e.g., assuming that politics respond directly to social concern), so that

$$P(SO \cap PO) = P(SO)P(PO/SO) \tag{4}$$

In the case that SO and PO are independent events, equation (4) simplifies to

$$P(SO \cap PO) = P(SO)P(PO) \tag{5}$$

This system is transferred to the evaluation of basic events in terms of those placed at a lesser level, and so on.

### 3.4 Computing the individual probabilities of events and the global system failure

The next step is to assign probabilities to all events in the bottom of the tree, and then build up (bottom-up) to assign probabilities using the Boolean rules above, until the top (full system failure) is reached. As a first step, all events were included into four categories depending on probability of occurrence (high, medium, low, or no risk).

A key point in the assessment of risk is assigning probabilities to each individual basic event, this process being quite challenging. One advantage of the fault tree approach is the possibility of assigning them at several stages of involvement, taking into account a combination of simplicity and relevance. The approach consists, first, in assigning a (preliminary) value to all basic events; these values may be based on the experience of the managers of the facility or experts. Such values are combined by the Boolean rules to provide a map of critical paths; i.e., events that are up in the tree and that result in high probability of failure.

The second step would be to devote attention to these significant events and the possibility to correct or reduce their risk in order to reduce the global risk. For those significant events, whose risk contribution is largest, probabilities can be reassigned by using sophisticated approaches, based for example on conceptual or numerical modeling, and also on changing sampling schemes or putting into operation new observation networks. From these new values, the critical events are re-assessed (including total failure). The process can be repeated as many times as needed, to arrive to an improved value of system failure.





In addition, corrective and preventive measures could be set to reduce the probability for individual events. The full procedure could go on until either economic resources are exhausted, or else it is considered that further refinement cannot lead to a significant improvement of the final figure.

We provide an example for the purpose of illustration. Let us assume a surface infiltration pond located in a flood plain. We can start by assigning some probability of the MAR facility being affected by flooding (meaning that operation would have to be discontinued for a long time) using a qualitative approach provided by the facility managers (high / intermediate / low / no risk), maybe including the input of local people that would tell us about potential flooding events that took place during their lifetime. It is very relevant to state that these preliminary numbers should always be on the conservative side, meaning that the less technical the evaluation is, the more caution should be included in the actual figures. The second step would use the idea that the facility is located in a 100 year flood plain; if we consider the life of the facility of 30 years, and from simple statistics, we can evaluate the probability that flooding occurs during the lifetime is $1-0.99^{30}=0.26$. Now, if this number is excessively high and relevant for the evaluation of overall failure, a further step may include a full hydrological analysis through modeling to re-evaluate the probability of flooding.

Additionally, prevention measures for reducing risk by using protection works such as embankments construction may be included. Such a measure could then affect indirectly some events (water quality, social acceptance, . . . ), leading to the need to continuously update all event risks. This step requires a deep knowledge of the system, and must be done under a local perspective and case by case. As the main goal of this paper is to develop a general methodology to evaluate MAR failure and to compare six sites, this type of analysis is out of scope, and we did not perform any detailed analysis of the risk reduction of any event.

## 4  Description of the field sites

The PTA-FT analysis was performed in six MAR facilities, located in different parts of the Mediterranean basin, offering a broad view of risk perception in the whole area. A summary of the characteristics and context of these sites can be found in Table 2.

The first site is located in the Algarve region (south of Portugal). It is based on an infiltration basin constructed in the Rio Seco river bed (Campina de Faro Aquifer system). This MAR facility is aimed at improving the water quality of the Campina de Faro aquifer. It was constructed in 2006 (Ferreira and Leitao, 2014). The surroundings of the MAR facility are mainly agricultural and one of its main problems is related to water quality due to agriculture diffuse water pollution, mainly by nitrate (Leitão, T., Lobo-Ferreira, J.P., Martins, T., Oliveira, M. M., Henriques, 2017). This site includes other MAR facilities in the Querença-Silves aquifer and Melides watershed, not included in this evaluation.

The second site, Los Arenales, is located in the center of Spain (Castilla y León). The MAR facility is aimed at providing enough water for the development of rural activities in the zone. Besides this, MAR is also aimed at improving the groundwater quality (reducing nitrate concentrations). In this case, the site is an ensemble of different small facilities (infiltration ponds,



river bank filtration and infiltration wells) located in the same aquifer (Los Arenales alluvial aquifer). It started its activity in 2002 and it was expanded in 2003, 2004 and in 2012.

The MAR Llobregat site is located in Catalonia (NorthEast Spain), in an alluvial aquifer placed some 10 km SW of Barcelona City. It is composed of one settling pond and one for infiltration. The recharge water comes from the Llobregat River and the

main goal of the facility is to increase the water storage in the aquifer, as well as to improve the quality of the recharged water. A reactive layer was placed at the bottom of the infiltration pond to improve the degradation of both traditional pollutants and emerging organic compounds (e.g. pharmaceuticals) (Valhondo et al., 2015). The area surrounding the facility involves agricultural, industrial and urban uses. It started its activity in 2009.

The induced riverbank filtration scheme along the Serchio River in Sant'Alessio is located in Lucca (Italy). The main goal of

this MAR facility is to provide continuous availability of water with good chemical quality for drinking uses to the people of the coastal Tuscany (Rossetto et al., 2015). The surroundings of the zone are mainly peri-urban/rural. This facility provides 15 Mm$^3$/year and started its activity in the '60s; it was further improved by building a river weir to increase storage at the beginning of 2000.

The Menashe site is located in Israel. Constructed in 1967, the site includes a settling pond and 3 infiltration ponds and a canal

in which storm water flowing in ephemeral streams are diverted for infiltration in sand dunes overlaying the northern part of the Israeli Coastal Aquifer. Since 2013, the facility is used also for infiltration of desalinated-seawater (1-3% of production) from the nearby Hadera desalination plant on the Mediterranean coast (Ganot et al., 2017; Ronen-Eliraz et al., 2017). Freshwater is recovered from wells surrounding the infiltration ponds mainly for residential and industrial consumers.

The Malta site is located in the South Malta Coastal Aquifer. The main objective of this site is the implementation of a MAR

facility to act as a seawater intrusion barrier and to minimize the salinization risk of the aquifer using a series of infiltration boreholes. The site is located on the coastal margin of a predominantly agricultural region in a limestone aquifer. The activity started in 2016, and is considered as a pilot site to guide the future implementation of MAR in the Maltese islands.

## 5   Probability assignment and global risk computation

### 5.1   Risk probability assignment

The probabilities for the basic events were defined by the personal in charge of each MAR facility, according to their own experience (expert criteria). These experts had to fill a simple questionnaire providing the four categories in terms of frequency of events (high, medium, low and no risk) for the sixty-five base risk events. These questionnaires followed the same schemes as the fault trees. The values provided by the experts reflected the probability that the MAR facility failed due to one of these

basic events during a period of 2-6 years. The qualitative answer were then translated to absolute values of probabilities, in coherence with the importance of the event in a potential failure of the facility.





## 5.2 Global risk computation: MAR-RISK APP

Once the probability values for each basic event were defined, and the questionnaires filled, global risk values for each facility were computed using a visual tool application, the MAR-RISKAPP. This tool was carried out in a friendly interface, aimed at being used by the managers of MAR facilities worldwide. The tool allows the user to assign one of the four risk categories to each basic event. A value of probability is then assigned by default to each event and category. The user can then manually modify each one of the probability values to keep updating the values of the full tree. The global probability of system failure is then computed internally.

The MAR-RISKAPP is an open application which can be downloaded from the website (http://marsol.eu/35-0-Results.html). The main flowchart of the APP is shown in Figure 4 and the manual of the app is summarized in S3 of supplementary material.

## 6 Results and Discussion

### 6.1 Comparison of risk probabilities defined by expert criteria

As a first step, we have compared the differences between the values provided by the facility managers, internally incorporating personal knowledge and technical expertise, in the six sites. Notice that this way we compare "perception of risk" rather than actual risk. The results, presented as a box plot of all the values reported by categories, showed that the larger values of risk perceived corresponded to events classified as non-technical (Figure 5). The risk values (in probability terms) in decreasing order were: Legal constraints, Social aspects, and Economic constraints. On the other hand, for the technical part, the order was (also in decreasing order): Water quantity, Structural damage, and Water quality. Therefore, the perception of risk of the managers of each MAR facility, based on their knowledge and experience, indicates that non-technical aspects are critical and can eventually lead to the facility failing to operate; it might imply that during operation and when the facility has been located in a technically appropriate site, much more uncertainty is expected to non-technical issues than to technical ones.

Legislation was the category with highest risk perception. In general terms, this is explained by lack or extremely new (such in Italy with DM 100/2016) pieces of specific legislation about managed aquifer recharge at the European level. The existing European water directives only provide little guidance for authorizing aquifer recharge schemes (Hochstrat et al., 2010). Consequently, MAR regulation is covered by different institutions and authorities dealing with water, environmental and health legislation. For example, in Menashe, the water recharged is to be used as drinking water, therefore the health legislation risk exceeds other associated legal risks. On the other hand, in Algarve, the infiltration zone is inside a nitrate vulnerable zone and a coastal nature reserve, leading to highest environmental regulations risks. In the Malta case, highest legal risks are associated to the potential for saline intrusion.

The following category in terms of risk perception was social aspects, related to the unacceptance of recharge technology by the society. We believe that this could be related, again, to the lack of a concise legislation, which creates social uncertainty. We observed that social issues were mostly present in those sites with strong political implications, where the public administrations or the agricultural users participate in the management (and even in the construction) of the MAR facility, such as





Llobregat, Los Arenales, or Malta.

Regarding the water quantity aspects, their relatively high importance could be explained because the sites are located in a Mediterranean climate (floods and droughts are typical in such environments). Besides this, it is related to the infiltration capacity of the system (especially in infiltration basin like Los Arenales and Llobregat) and this is traditionally one of the main technical issues in MAR (Figure 1). Nevertheless, in general terms, infiltration capacity of the system was not an important category in risk perception, probably because the sites were located in high permeable zones, suitable for recharge, and most of the sites included in their maintenance tasks actions to minimize its importance.

The risk perception on the quality issues could include the next topics: recovered-water use and water source. For example, Serchio recharged water is used as a drinking water, consequently, quality plays a higher role than, for example, in Malta, where water is used as a water barrier to salt intrusion. On the other hand, quality is also important in sites where quality aspects existed independently of recharge, like Arenales (high presence of nitrate in groundwater due to farming activity in the zone, see San Sebastián et al., 2017) and Llobregat (quality problems related to industrial and urban activities, see Valhondo et al. (2015).

Structural Damages category is non-negligible, but in general it is not perceived as critical, probably because we are considering sites already in operation. This issue could be more significant in the design process of a facility, and also during construction.

## 6.2 Comparing risk in the different sites

The risk values for the six sites studied are summarized in Figure 6. We can observe that the total risk of three of the six sites (Los Arenales, Algarve and Llobregat) is very large (equal or above 0.9), indicating that facility failure is almost certain during a 2-6 year period. This indicates that the system will most probably have to discontinue operation; however, this does not imply that the system cannot be put back to operation again. Lack of specific legislation, economic constraints, social issues, and quantitative aspects are the most probable cause for failure of these three facilities. Regarding water quantity being a potential cause of failure, the fact that all three sites involve surface infiltration with river water promote that quantity and clogging aspects are important, as Mediterranean rivers display low flows and high solid content. Besides this, these three sites are quite young (around 10 years old), which could imply that are not completely optimized.

On the other hand, Serchio, Menashe, and Malta have lower risk values (0.18, 0.29, and 0.75 respectively). The cases of Serchio and Menashe can be explained because they are the oldest sites, with large experience in the operation of the facilities (therefore lowering the Technical risks to below 0.1). Furthermore, river bank filtration in Serchio is done with water with less suspended solids, so that the risk of clogging is low. The case of Menashe is justified by the use of desalinated and storm water for recharge. The presence of solids in these two recharge waters is very low. In the case of Malta, the low risk value could be just perception based on the site just recently started operation.

From all technical constrains, the one with the highest risk is related with water quantity. Half of the sites showed significant risk in terms of quantity, somewhat correlated with the sources of water for infiltration (so, being largest for those relying on surface water, Algarve, Llobregat, and Arenales). In the case of Malta, quantitative problems are related to the need to produce good quality water from wastewater. Algarve site is a particular case as its aim is to improve groundwater quality





with recharging water from a non-perennial stream (surface water flows only 60-70 days per year). So, the lack of water is already considered in the MAR scheme. In terms of quality, again, the three sites supplied with river water are those showing the highest risks. That could respond to the variability of river water quality along the year. The geological/hydrogeological context does not seem to have any effect in the technical risk values despite it is very significant to define the site were to

construct the MAR facilities.

Individually, the main risk issues for the technical issues in the Llobregat site were quantitative aspects, mainly clogging due to fine particles (probability = 0.4) and recharge water turbidity (0.4). The non-technical issues, were mostly related to social aspects: lack of coordination amongst stake holders (0.4), children surveillance (0.3), and fair distribution of water (0.3). These three social aspects are aligned with the indicators of acceptance of Mankad and Walton (2015). For the Algarve

site, the technical issues were mostly potential flooding (0.3), droughts (0.3), vandalism/terrorism (0.2) and clogging (0.2). Non-technical issues were mainly related to regional/local legislation (0.3). Non-technical issues for the Menashe site include domestic water use (0.15), perception of effectiveness (0.05) and high cost perception (0.05); for the technical ones, terrorism activities (0.02) and clogging due to compaction (0.02) were the most significant.

Serchio site had for the non-technical main risks the lack of knowledge on MAR activities (0.05) and health legislation (0.01).

About the Technical aspects, quality aspects related to organic compounds were the largest (0.01). Los Arenales site had very large risk values associated with national (0.6), regional/local (0.8) and other legislation (0.5), agricultural water use (0.6), and fair distribution of water (0.45); the most significant technical issues were flooding (0.1), nutrients in the recharge water (0.5), droughts (0.8), generation of gas - physical clogging (0.2), nitrogen metabolites (0.2), river (0.1) and wetland water levels (0.2). Malta site identified the European legislation (0.2) and lack of coordination (0.1) as the main non-technical risk drivers.

About the technical aspects largest risks included pipe breakage (0.05), and different specific targets: seawater barriers (0.4), protected water body (0.1) and groundwater levels (0.3).

The risk values obtained are mostly correlated to the expert criteria values. This correlation was evaluated by applying a Pearson product-moment correlation between the Expert criteria basic events (considered the Medium Risk values) and the Results basic events for each site. There were in total 65 basic events, leaving a total of 63 degrees of freedom and considering

a p-value of 0.05 as the confidence limit of acceptance. It was observed that, in general terms, a correlation between the perception of risk (Expert criteria) and actual risk (Results) existed. This mainly means that the facility managers know the main problems of the sites and thus define the Expert criteria values accordingly. This indicates the relevance of using such a simplified method for preliminary risk assessment.

The actual results of the analyses showed correlation (in terms of p-value) in the cases of Llobregat (p=0.026), Los Arenales

($7.38 \times 10^{-13}$), Malta ($3.2 10^{-7}$), Algarve (0.048) and Serchio ($2.12 \times 10^{-8}$), while for Menashe site (0.6), that correlation could not be observed. Looking at the data, Menashe Expert criteria values lack absolute zero values (0 from 65), however the result values obtained show a high proportion of NO RISK (risk = 0) values (52 from the total of 65). Then the difference between both Expert criteria and Results become apparent, probably related to the knowledge of the personnel in charge of the Menashe site and their confidence in the lack of risk of their operations.



## 7   Conclusions

In this paper, we have developed a methodology to evaluate the risk of failure of Managed Aquifer Recharge (MAR) facilities, and we have applied it to six different facilities located in the Mediterranean Basin. The methodology was based on the development of a Probabilistic Risk Assessment based on Fault Trees. The PRA-FT methodology considered different categories
affecting the operation of the facility. We further defined 65 basic events that individually or properly combined can produce global failure of the MAR facility. These events were compiled from a literature review of 51 MAR facilities and, then, extended with the results of the European Project MARSOL ("Demonstrating Managed Aquifer Recharge as a Solution to Water Scarcity and Drought").

The methodology consists of providing probability values to all basic events to take place in a window of time. Then, event
at an upper level are computed from Boolean Algebra until the top of the tree (total failure) is quantified. The initial step is to provide values based on Expert Criteria, assigned from the four risk categories: low, medium, high, and no risk. All values can be updated sequentially and probabilities are recalculated, until the values converge. The basic events include both technical and non-technical events.

A full preliminary (without updating) assessment of risk was developed for 6 sites located in the Mediterranean Basin. It was
found that the non-technical aspects can be the most significant ones, contributing more than the technical issues to the overall assessment of risk. This is despite we are considering only facilities under operation, so that some issues are supposed to be already resolved. In short, the combination of legal and economic factors can be really a strong contribution to global risk.

All events considered, we found that in the facilities analyzed, the major contributors to overall risk were, in decreasing order of importance: Legal constraints, Social aspects, Economic constraints, Quantity issues, Structural damages, Specific targets
and Quality issues. In particular, when the recharge water is supplied by a river, quantity aspects increase their relevance, due to the uncertainty in the future potential capacity for supplying in a dry and variable climate such as the Mediterranean Area.

The site-specific results were obtained from a questionnaire, and so they provide "perception of risk" rather than "actual risk", and thus could and should be amended. The PRA-FT methodology allows now to concentrate on the specific issues that individually, or combined, lead to the largest probability of failure, and concentrate the efforts in updating such values by means
of detailed evaluations or specific projects of rehabilitation. The system can go on using any number of reevaluations until an acceptable value or either until no further improvement can be obtained.

Regarding the results on perception of risk for the individual sites, it was surprising to get three of them (Los Arenales, Algarve, and Llobregat) above 0.90 in a 2-6 y period. The main contributors to failure were related to non-technical reasons and to quantity aspects. Actually, in recent years all three facilities had to discontinue operations at least one, indicating that the
evaluations provide reasonable estimations. The Malta site is a very recent one, with little history behind, and this it is not possible to evaluate whether the perception of risk of 75% is high or low.

On the other hand, the risks perceived for the other two sites, Serchio and Menashe, can be considered low (18% and 29%, respectively). A potential reason is that they are the oldest facilities, so that experience has been accumulated for decades. Also, the facilities have been able to adapt to evolving regulations (both local and at the European levels). In both cases, low



risk values correlated also with low perception of quantitative risk (mainly related to clogging), due to the sources of recharge

water in both cases (river bank filtration in Serchio and desalinated and storm water in Menashe.

*Competing interests.* The authors declare that they have no conflict of interest

*Acknowledgements.* Financial support was provided by the project MARSOL grant agreement no. 619120, FP7-ENV-2013-WATER-INNO-
5    DEMO, by the European Union, project ACWAPUR, PCIN-2015-239, project INDEMNE, CGL2015-69768-R (MINECO/FEDER) by the

Spanish government. All data is available from the corresponding author upon request.



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





**Figure 1.** Sorted list in terms of frequency in appearance of the main problems observed in reviewed facilities of deep well injection and infiltration basin. Problems are classified into categories (five for technical, three for non-technical) that are visualized as colors.







**Figure 2.** General fault tree for the operational phase.





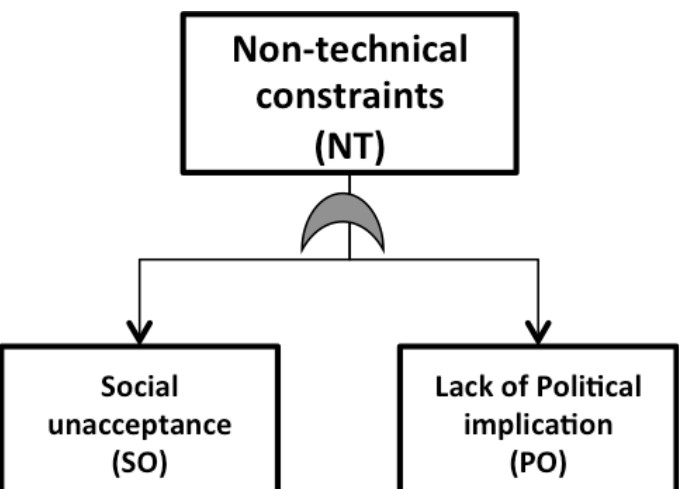

**Figure 3.** Simplified illustrative case for Non-technical constraints involving only sociopolitical constraints. The symbol below the upper event represents and "OR" operator.



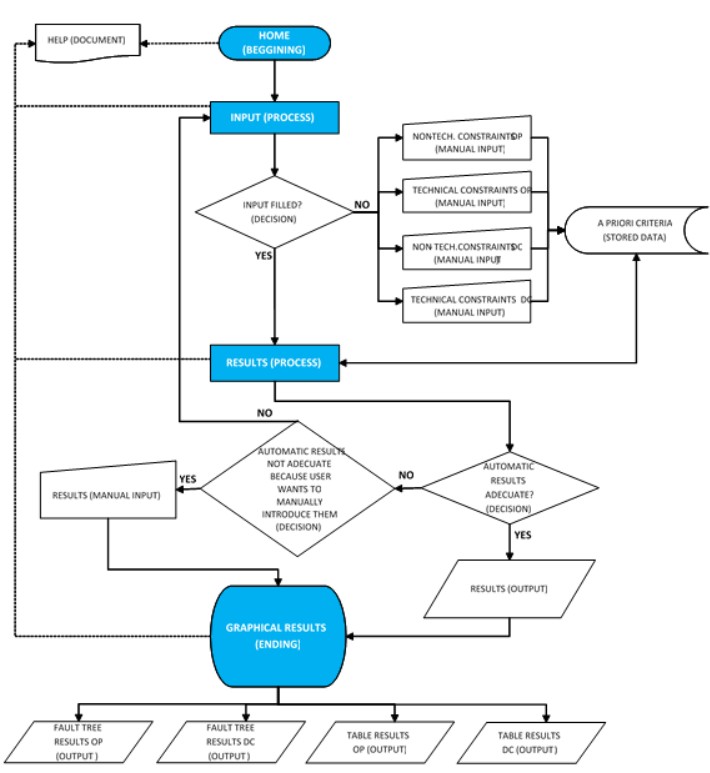

**Figure 4.** Flowchart for the main program of the MAR-RISKAPP.



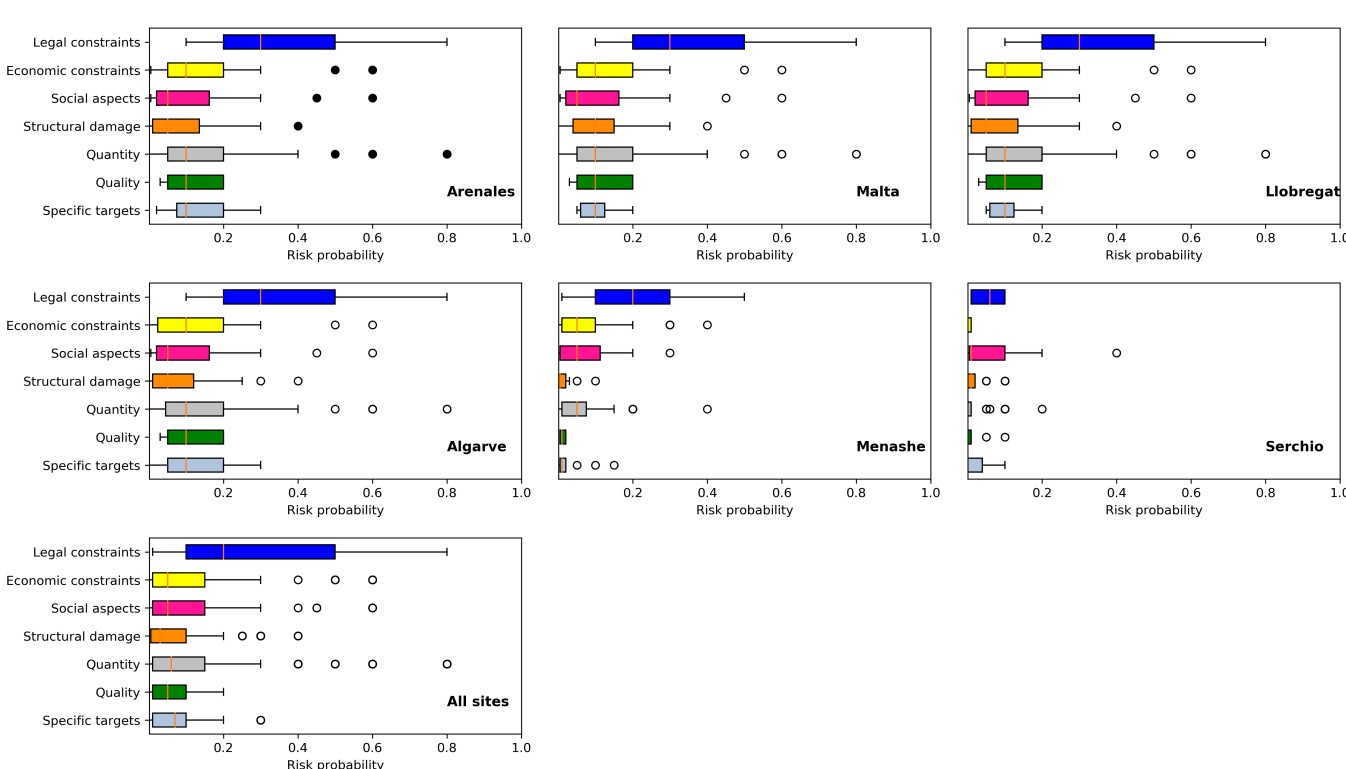

**Figure 5.** Distribution of the expert criteria by category and for the three levels (low, medium, high) grouped by categories.





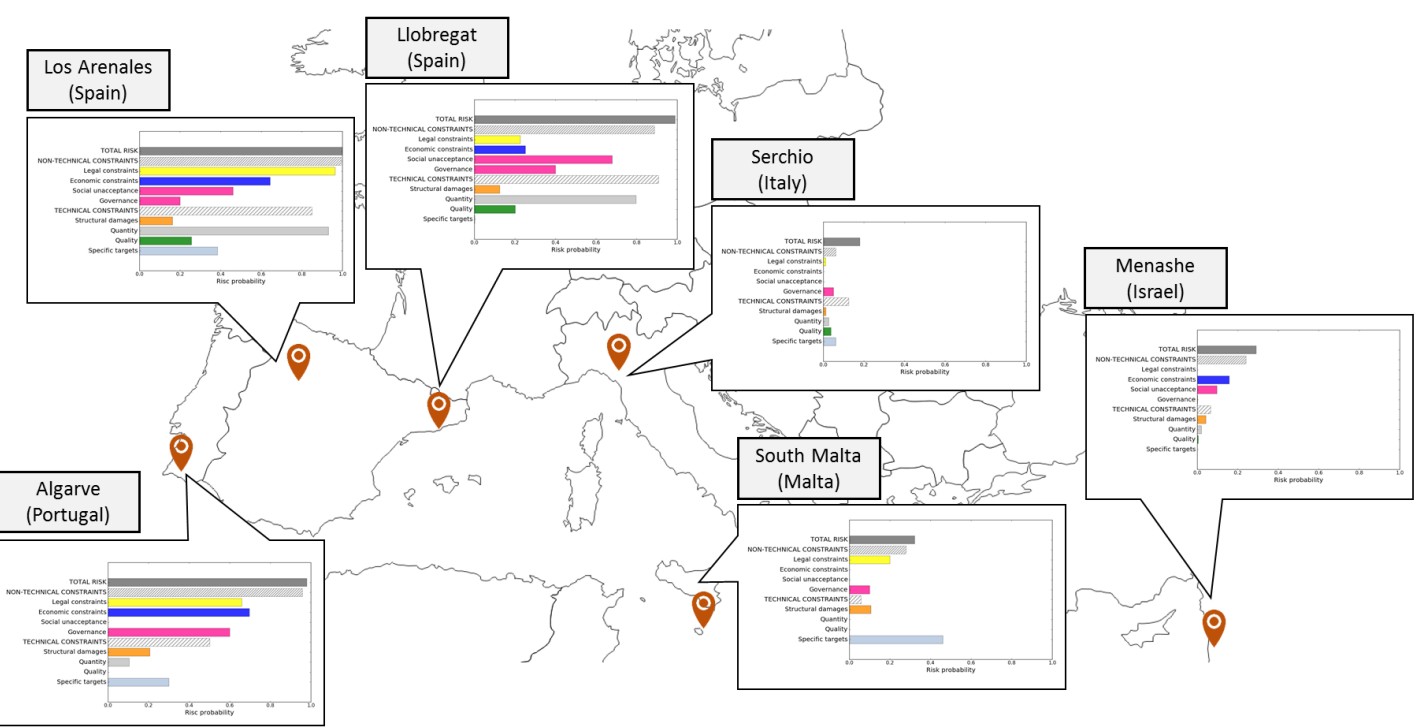

**Figure 6.** Risk in the different MAR sites.





**Table 1. Events of the fault tree divided by categories for non-technical and technical issues**

| | |
|---|---|
| Non-Technical | Legal constraints (LEG): health, urban, environment, construction permits |
| | Economic constraints (ECO): lack of funds, maintenance/installation costs, macro and microeconomical problems... |
| | Social unacceptance (SO): health perception, cost perception, effectiveness perception... |
| | Governance (GOV): coordination between governmental agencies and technical knowledge about the MAR issues |
| Technical | Structural damage (SD): damage to the MAR infrastructure due to natural hazards, civil works failure, etc. |
| | Not enough water or Quantity (QUAT): low water quality (physical, chemical and biological), water scarcity (climate, river regulation, WWTP failure, quantity recharged does not reach some target value that makes it economically feasible ) and clogging (physical, biological and chemical) water available does not reach the quality standards needed to allow it to be used in the recharge facility). |
| | Unacceptable water quality (QUAL): problems with natural attenuation (nutrients, organic matter and emerging organic compounds), metabolites (nitrogen cycle, other nutrients like $H_2S$, etc.) and mobilization of metals. The water finally resulting in the aquifer does not meet some quality standards once it reaches some sensitive location (river, supply well, wetland, ...). |
| | Specific targets (ST): failure to achieve targets related to seawater barriers, protected water bodies and water levels. Seawater intrusion is not sufficiently contained, a protected water body is reached by polluted water or water levels at target surface water bodies (river, spring, wetland) are not reached. |





**Table 2.** Events of the fault tree divided by categories for non-technical and technical issues

|  | **Algarve** | **Los Arenales** | **Llobregat** | **Serchio** | **Menashe** | **Malta** |
|---|---|---|---|---|---|---|
| **Location** | Algarve (Portugal) | Los Arenales (Spain) | Sant Vicenç dels Horts (Spain) | Serchio (Italy) | Menashe (Israel) | South Malta (Malta) |
| **Type of recharge** | Surface infiltration basins and large wells | Surface infiltration (channels, ditches, ponds) and wells | Surface infiltration basin | Riverbank filtration | Surface infiltration basin | Deep wells |
| **Source of recharge water** | River water and WWTP water | River water | River water | River water | Storm water and desalinated seawater | WWTP water |
| **Use of recharged water** | Improving aquifer water quality and aquifer storage to prevent seawater intrusion | Agriculture | Improve aquifer quantity and quality | Improve aquifer quantity and quality | Store excess of storm and desalinated water | Coastal barrier for seawater intrusion, increase water quantity and quality |
| **Surrounding** | Farmland | Farmland | Farmland and industrial park | Coastal zone and industrial areas | Farmland, industrial and urban areas | Coastal zone, farmland |
| **Aquifer geology** | Alluvial | Aeolian sandy | Alluvial | Sand and gravel alluvial | Interlayered sands calcareous-sandstone and clays | Coastal, floating-lens aquifer |
| **Political support** | guas do Algarve | Spanish Ministry of Agriculture, Fishing, Food and Environment | Catalan Water Agency | Provincia di Lucca administration | Mekorot National Water Company | Malta Reources Authority and Water Services Corporation |
| **Social setup** | Farmers irrigation associations willing to contribute to financing MAR | Farmers, small industry presence and local public administrations | Water Users Community (Farmers and industry presence) | NA | Pressure on land-use from industrial sector versus water sector | NA |