# Peer review of "S1. LITERARURE REVIEW OF MANAGED AQUIFER RECHARGE FAILURES"

_Hydrology and Earth System Sciences, 2018_

## Referee Comment (RC1) · A. Dell'Oca (Referee) · 7 Feb 2018

General Comment: The paper proposes an interesting application of Fault Trees and Probabilistic Risk Assessment methodologies to evaluate the risk of failure (i.e. need to stop operation facility) for Managed Aquifer Recharge (MAR). As novel aspects (at least to me) there are (i) the integration of both technical and non-technical aspects that could lead to failure and (ii) the quantification of the probability of such events on the base of experts/managers opinions. This last point could be controversial since, as correctly stated by the Authors at line 13 at pp 9, doing so the analysis is based on the 'perception of risk' rather than the 'actual risk'. I leave to the Editor the task of

judging if such approach is 'acceptable' or not, while I have really appreciated it and the related results and conclusions. Moreover, note that the Authors precisely state that their approach is a dynamical one, where as a first step critical combinations of event are highlighted in the Fault Trees and, eventually, further investigation for such events are conduct with the aim of investigating the 'actual risk'. Furthermore, the definition of the 'actual risk' of each event in the proposed Fault Trees (65 events both technical and non-technical) could be really challenging! That is way I appreciate the practical cut of this paper. I would recommend the publication after the Authors address some minor comments listed below.

Comment 1: Would it be possible to evaluate the MAR' failure probability on the base of the results in Fig. 1? I intend that frequency of events (based on the literature review and not the managers opinions) as the probabilities of each event 'unconditional' from the specific manager opinions. Maybe these frequencies/probabilities are more general and less conditioned by the manager point of view. Interestingly enough from Fig. 1 it seems that technical aspects are more frequent than the non-technical ones, while the analysis based on the managers opinions suggest an opposite behavior! (does bureaucracy and legislation bother the technician/managers more than technical problems?) I suggest the Authors to perform the analysis on the base of Fig.1- frequencies if they think I would be of interest.

Comment 2: In the introduction the Authors state that in Probabilistic Risk Assessment (PRA) the risk is defined 'as the probability of an undesired outcome to happen (evaluated in percentage of occurrence, return period, etc.) and an evaluation of the potential damage that a particular outcome might cause (amount of damage, adverse hetalth effects, etc.)'. I agree with this definition, but in the rest of the work it seem to me that the Authors deal only with first part, i.e. definition of probabilities of failure, and not with the evaluation of the potential damage, that would lead to the risk. I suggest to add a statement about this point or to review the terminology trough out the paper since, to me, it seems that probability of failure is the right wording instead of risk of failure.

Note that in Section 3.3 the Authors only deal with the probabilistic representation of the fault tree and no mention is done to the evaluation/representation of the potential damage.

Comment 3: At line 9-10 at pp. 7 the Authors state 'it is very relevant to state that these preliminary numbers (i.e. probability of events assigned by managers or local people opinions) should always be on the conservative side, meaning that the less technical the evaluation is, the more caution should be included in the actual figures'. I agree with this statement, but immediately started to wonder 'which is the conservative side? Is it the high probability of failure of the MAR facility (e.g. as wished by some local people) or is it the low probability of failure (e.g. as wished by investors)?'. What do the Authors think about this point?

Comment 4: I really appreciate the description in Section 3.4 of the dynamicity of the proposed approach!

Comment 5: Section 5.1 risk probability assignment. From the provided text is not clear to me how 'the qualitative answers were then translated to absolute values of probabilities, in coherence with the importance of the event in a potential failure of the facility'. Could the Authors elaborate more on the way they assigned probabilities in the fault tree starting from the experts' opinions? This point could be useful and relevant to interested readers.

Comment 6: Please increase the quality of the writing part in Fig.2, Fig. 4 and Fig. 6. It is really hard to read them.

---

## Referee Comment (RC2) · G. Ghiglieri (Referee) · 18 Apr 2018

Review of HESS paper entitled: A risk assessment methodology to evaluate the risk failure of Managed Aquifer Recharge in Mediterranean basin By: Paula Rodríguez-Escales et al. DOI: https://doi.org/10.5194/hess-2018-8

This paper discusses about a risk assessment methodology to evaluate the risk failure of Managed Aquifer Recharge in Mediterranean basin. The authors have applied this methodology to six different Managed Aquifer Recharge sites located at the Mediterranean Basin (Portugal, Spain, Italy, Malta and Israel). The probabilities of the basic events were defined by expert criteria, based on the knowledge of the different man-

agers of the facilities. The paper is concise, correctly organized and their discussions are sound and convincing. This paper has a high scientific level because of the amount of data presented, the complexity of the discussion and, last but not least, its potential application to water resources management. The paper deals with a topic which is of worldwide concern. For this reason, I think that any contribution to the above topic has to be considered as important by scientific point of view. This last point has an interest because professionals and public institutions draw from the scientific literature the methods that they will apply to real cases, and these methods become often standard code of practice.

The paper is well-written, well-referenced and well-structured: accepted Giorgio Ghiglieri

---

## Referee Comment (RC3) · Y. Kim (Referee) · 21 Apr 2018

General Comments

This paper is presenting a risk assessment methodology to evaluate the risk failure of six MAR sites in Mediterranean basin using PRA-FT and highly recommend to publish with minor revision.

Specific Comments

As a result and conclusion, it is stated that non-technical factors such as legal constraint due to lack of legislation, social aspects and economic constraints are most sig-

nificant ones contributing more than the technical issues to the overall risk assessment. This means, I think, the technical factors in quantity and quality have been studied and solved in many scientific research efforts. So I would suggest the authors to include the necessity and importance of future works to lower the risks by the non-technical factors to make MAR methods to be effective solution for water issues in the end of conclusion.

Technical Corrections

Page 11 line 30- typo: Malta (3.210-7)

Page 12 line 30 – typo: and this it is not

Page 20 Fig 2. – mis-match between LQIP in diagram and LWIP in legend.
* * *

---

## Author Comment (AC2) · 30 Apr 2018

We really thank Dr. Ghiglieri for the nice comments. We are glad to see that our work has been this nice acceptance. We also hope our work will be useful for professionals and public institutions to improve water management.
* * *

---

## Author Comment (AC3) · 30 Apr 2018

GENERAL COMMENTS: This paper is presenting a risk assessment methodology to evaluate the risk failure of six MAR sites in Mediterranean basin using PRA-FT and highly recommend to publish with minor revision.

RESPONSE: We thank Dr. YongCheol Kim for his comments and suggestions. In the spirit of HESS discussions, we discuss below the issues that are potentially controversial or that require further explanations. Editorial corrections will be included in the revised manuscript, when a full response to all reviewer comments will be produced, but are not addressed at this point.

[Figure]

COMMENT 1: As a result and conclusion, it is stated that non-technical factors such as legal constraint due to lack of legislation, social aspects and economic constraints are most significant ones contributing more than the technical issues to the overall risk assessment. This means, I think, the technical factors in quantity and quality have been studied and solved in many scientific research efforts. So I would suggest the authors to include the necessity and importance of future works to lower the risks by the non-technical factors to make MAR methods to be effective solution for water issues in the end of conclusion.

RESPONSE: We agree with the reviewer. We will highlight in the Conclusions that further research in non-technical aspects is needed to lower the risks in Managed Aquifer Recharge.

We will also include all the technical corrections in the last version of the manuscript.

---

## Author Response (AR1)

**ANSWER TO REVIEWERS**

We hereby present the detailed answers to all the comments provided by the reviewers. We sincerely hope that the answers clarify the different points and that the main changes included in the revised version are sufficiently highlighted.

To facilitate the revision, the answers to the different comments are written in blue font right (and in *italic*) under the corresponding original comment (in black). In the manuscript, the modifications are shown in red.

**Reply to comments of Dr. Dell'Oca:**

**General Comment:** The paper proposes an interesting application of Fault Trees and Probabilistic Risk Assessment methodologies to evaluate the risk of failure (i.e. need to stop operation facility) for Managed Aquifer Recharge (MAR). Novel aspects (at least to me) are (i) the integration of both technical and non-technical aspects that could lead to failure and (ii) the quantification of the probability of such events on the base of experts/managers opinions. This last point could be controversial since, as correctly stated by the Authors at line 13 at pp 9, doing so the analysis is based on the 'perception of risk' rather than the 'actual risk'. I leave to the Editor the task of judging if such approach is 'acceptable' or not, while I have really appreciated it and the related results and conclusions. Moreover, note that the Authors precisely state that their approach is a dynamical one, where as a first step critical combinations of event are highlighted in the Fault Trees and, eventually, further investigation for such events are conduct with the aim of investigating the 'actual risk'. Furthermore, the definition of the 'actual risk' of each event in the proposed Fault Trees (65 events both technical and non-technical) could be really challenging! That is why I appreciate the practical cut of this paper. I would recommend the publication after the Authors address some minor comments listed below.

*We thank Dr. Dell'Oca for his comments and suggestions. We discuss below the issues that are potentially controversial or that require further explanations.*

**Comment 1:** Would it be possible to evaluate the MAR' failure probability on the base of the results in Fig. 1? I intend that frequency of events (based on the literature review and not the managers' opinions) as the probabilities of each event 'unconditional' from the specific managers' opinions. Maybe these frequencies/probabilities are more general and less conditioned by the manager point of view. Interestingly enough from Fig. 1 it seems that technical aspects are more frequent than the non-technical ones, while the analysis based on the managers opinions suggest an opposite behavior! (Do bureaucracy and legislation bother

the technician/managers more than technical problems?) I suggest the Authors to perform the analysis on the base of Fig.1- frequencies if they think it would be of interest.

*This is a good point, but the answer is: in general, no. We used data from Fig 1 to produce some initial guesses that could be used for a new facility if no additional input was involved. Now, MAR facilities are all quite distinct from each other, and the national and local realities are really very significant. An evaluation based on data from Fig 1 would be of little relevance. We understand the scientific point of view of the reviewer, but here conditioning on expert opinions and local geological and hydrological conditions make unconditional guess quite irrelevant.*

*We believe that one of the challenging aspects of our work is to evaluate and to compare the risk of six Mediterranean MAR sites considering the 65 quasi-independent events. We believe that developing the analysis from the events defined in the literature (supporting information and Figure 1) could be a little tendentious, since we only know what the report/paper described. Furthermore, the different events are defined in a different way in the different sites. Thus, we can only define categories and not basic events. Then, we relied on additional support material, plus the authors' experience and the results from the EU project to elucidate the basic events that were finally included in the tree.*

*There is an additional point that emerges from the referee's comment. As he correctly noticed, the most significant events that we found in the literature were the technical ones, rather than the non-technical. Then our analysis on the six Mediterranean sites concluded the opposite. We do not have enough information to know if this is real, but we really think so, and we attribute this discrepancy in a clear bias in the scientific literature to technical issues. We could not find information on MAR facilities failure in journals devoted to social or economic sciences. Independently of this, we believe is quite interesting to add this difference in the discussion of the paper. So, we have improved section 6.2 of the manuscript adding a discussion based on the comparison of the Figure 1 and the current results of the MAR RISKAPP.*

> **LINES 1-5, PAGE 12**
>
> "A significant point to make is that there is a discrepancy between the literature review and our results. The most significant events leading to failure that we found in the literature were the technical ones, rather than the non-technical (recall Figure 1). Then our analysis on the six Mediterranean sites concluded the opposite. We attribute this discrepancy to a bias in the scientific literature towards technical issues. We could not

find information on MAR facilities failure in journals devoted to social or economic

sciences."

**Comment 2:** In the introduction the Authors state that in Probabilistic Risk Assessment (PRA) the risk is defined 'as the probability of an undesired outcome to happen (evaluated in percentage of occurrence, return period, etc.) and an evaluation of the potential damage that a particular outcome might cause (amount of damage, adverse health effects, etc.)'. I agree with this definition, but in the rest of the work it seem to me that the Authors deal only with first part, i.e. definition of probabilities of failure, and not with the evaluation of the potential damage, that would lead to the risk. I suggest to add a statement about this point or to review the terminology trough out the paper since, to me, it seems that probability of failure is the right wording instead of risk of failure.

Note that in Section 3.3 the Authors only deal with the probabilistic representation of the fault tree and no mention is done to the evaluation/representation of the potential damage.

*We agree with the reviewer and we have improved the definition according to the use we have done. As commented by the reviewer, our work has been focused on the probability of an undesired outcome to happen (the MAR facility ceasing operation) instead of the evaluation of the potential damage caused.*

*We have improved the definition in the final revised manuscript. The modifications have been done in the definition in the Section 1 and a complete revision of the text was done to ensure the complete text will be consistent.*

> **LINES 9-13, PAGE 3:**
>
> "Risk is defined here as the probability of an undesired outcome to happen (evaluated in terms of percentage of occurrence) which causes a damage (directly i.e. breakage of pipes or indirectly i.e. perception of effectiveness) to the recharge system and therefore causes a failure during operation or design of the MAR facility."

**Comment 3:** At line 9-10 at pp. 7 the Authors state 'it is very relevant to state that these preliminary numbers (i.e. probability of events assigned by managers or local people opinions) should always be on the conservative side, meaning that the less technical the evaluation is, the more caution should be included in the actual figures'. I agree with this statement, but immediately started to wonder 'which is the conservative side? Is it the high probability of failure

of the MAR facility (e.g. as wished by some local people) or is it the low probability of failure (e.g. as wished by investors)?'. What do the Authors think about this point?

*It is quite clear that most of the basic events can be considered uncertain. Therefore, in most of them there is a lack of specific knowledge of the actual risk values associated to the basic events, and this uncertainty increases whenever the evaluation is less technical and more based on opinions or data from the literature. In such a case, we want to use always an engineering approach, providing values that are on the conservative side. Now, the reviewer questions whether a conservative assessment would mean associating larger or smaller probabilities. This is correct, and it really depends on the target of the risk assessment. Nonetheless, we think that the way we posed the problem it should be clear that we are looking from either the administration of the managerial side, looking to potential problems and trying to improve the methodologies to reduce potential failure. So, in short to answer the reviewer, we consider "safe" to use values that are equal or above the actual probability of failure for the individual events.*

*We introduced this idea in the revised text (Section 3.4). Thanks for pointing it out.*

**LINES 9-12, PAGE 7:**

*"Here we consider the conservative side as the one providing a larger value of probability of failure than the real one (i.e., from the administration/manager point of view)."*

Comment 4: I really appreciate the description in Section 3.4 of the dynamicity of the proposed approach!

*Thanks for the nice comment.*

**Comment 5:** Section 5.1 risk probability assignment. From the provided text is not clear to me how 'the qualitative answers were then translated to absolute values of probabilities, in coherence with the importance of the event in a potential failure of the facility'. Could the Authors elaborate more on the way they assigned probabilities in the fault tree starting from the experts' opinions? This point could be useful and relevant to interested readers.

*We try to clarify it. The persons who answered the questionnaire had to choose between four different categorical variables (high, medium, low and no risk). The questionnaire includes the same basic events as the fault tree does, but sorted as a list at different levels (Event 1, Event 1.1., ...). At this point, we obtained some qualitative perception of risk for the different events. A default table associates these qualitative opinions into quantitative probability values (obviously*

*sorted from high to low); such values are given by default, but can be edited manually and updated whenever a better evaluation of probability is available (this is why we say that the values can be updated at any time, whenever new information becomes available). From then on, these values are included in the tool, and probability of events located at higher levels are computed based on the rules from Boolean Algebra, until the top box is reached.*

*We have also clarified this point in the manuscript in order to facilitate the global understanding of the methodology (Section 5.1).*

> **LINES 27-33. PAGE 8.**
>
> "The probabilities for the basic events were defined by the personal in charge of each MAR facility, according to their own experience (Expert Criteria). These values reflected the probability that the MAR facility failed due to the occurrence of one of these basic events, considering a time period of approximately 2-6 years. The Expert Criteria included values of each risk category (High, Medium and Low Risk) for each fault tree basic event. Then the users of the MAR-RISKAPP tool filled a qualitative questionnaire were for each basic event one of the four risk categories had to be chosen. This questionnaire reproduced the same events of the fault tree but sorted as a list at different levels. Then, the qualitative answers were translated to absolute values of probabilities using the values of the Expert Criteria and the rules of Boolean Algebras."

**Comment 6:** Please increase the quality of the writing part in Fig.2, Fig. 4 and Fig. 6. It is really hard to read them.

*Thanks for the comment. We have improved the legibility (and the quality) of the figures in the final revised manuscript.*

**Reply to comments of Dr. G. Ghiglieri:**

Review of HESS paper entitled: *"A risk assessment methodology to evaluate the risk failure of Managed Aquifer Recharge in Mediterranean basin"* By: Paula Rodríguez-Escales et al. DOI: https://doi.org/10.5194/hess-2018-8.

This paper discusses about a risk assessment methodology to evaluate the risk failure of Managed Aquifer Recharge in Mediterranean basin. The authors have applied this methodology to six different Managed Aquifer Recharge sites located at the Mediterranean Basin (Portugal, Spain, Italy, Malta and Israel). The probabilities of the basic events were defined by expert criteria, based on the knowledge of the different managers of the facilities. The paper is concise, correctly organized and their discussions are sound and convincing. This paper has a high scientific level because of the amount of data presented, the complexity of the discussion and, last but not least, its potential application to water resources management. The paper deals with a topic which is of worldwide concern. For this reason, I think that any contribution to the above topic has to be considered as important by scientific point of view. This last point has an interest because professionals and public institutions draw from the scientific literature the methods that they will apply to real cases, and these methods become often standard code of practice. The paper is well-written, well-referenced and well-structured: accepted Giorgio Ghiglieri

*We really thank to Dr. Ghiglieri the nice comments of our manuscript. We are glad to see that our work has been this nice acceptance. We also hope our work will be useful for professionals and public institutions to improve water management.*

**Reply to comments of Dr. Y. Kim:**

General Comments

This paper is presenting a risk assessment methodology to evaluate the risk failure of six MAR sites in Mediterranean basin using PRA-FT and highly recommend to publish with minor revision.

*We thank Dr. Y. Kim for his recommendation.*

Specific Comments

As a result and conclusion, it is stated that non-technical factors such as legal constraint due to lack of legislation, social aspects and economic constraints are most significant ones contributing more than the technical issues to the overall risk assessment. This means, I think, the technical factors in quantity and quality have been studied and solved in many scientific research efforts. So I would suggest the authors to include the necessity and importance of future works to lower the risks by the non-technical factors to make MAR methods to be effective solution for water issues in the end of conclusion.

*We agree with the reviewer and we have highlighted his suggestion in the conclusions of the paper.*

> **LINES 23-24. PAGE 12.**
>
> "Consequently, future risk works based in Managed Aquifer Recharge should consider how to lower risks by the non-technical factors."

Technical Corrections

Page 11 line 30- typo: Malta (3.210-7)

Done

Page 12 line 30 – typo: and this it is not

Done

Page 20 Fig 2. – missmatch between LQIP in diagram and LWIP in legend.

Done

[revised manuscript text omitted]